# Comparative analysis of diagnostic methods for the detection of *Cryptococcus neoformans* meningitis

**Katia Cristina Dantas**[1]*, **Roseli Santos de Freitas—Xavier**[2], **Suzete Cleusa Ferreira Spina Lombardi**[3], **Alfredo Mendroni Júnior**[4], **Marcos Vinicius da Silva**[5], **Paulo Ricardo Criado**[6], **Vera Lúcia Teixeira de Freitas**[7], **Terezinha Morato Bastos de Almeida**[8]

1 Department of Pathology, Sao Paulo University Medical School, Sao Paulo, Brazil, 2 Medical Mycology Laboratory-LIM 53/HCFMUSP and Institute of Tropical Medicine, University of Sao Paulo, São Paulo, Brazil, 3 Division of Research & Transfusion Medicine, Pro-Blood Foundation/Blood Center of São Paulo, Sao Paulo, Brazil, 4 Laboratory of Medical Investigation in Pathogenesis and Targeted Therapy in OncoImmuno-Hematology (LIM-31), Department of Hematology, Hospital das Clínicas -HCFMUSP, Faculty of Medicine, University of São Paulo, São Paulo, Brazil, 5 Emilio Ribas Institute of Infectious Diseases, Consultant, Ministry of Health, Department of Medicine, Catholic University of Sao Paulo, and Professor, Program in Postgraduate Sciences and Coordination of Disease Control, Department of State Health, São Paulo, Brazil, 6 Full Researcher at ABC Medical School, São Paulo, Brazil, 7 Laboratory of Medical Investigation in Imunology (LIM-48), Department of Infectious and Parasitic Diseases, Sao Paulo University Medical School, Sao Paulo, Brazil, 8 Department of Oncology, Radiology and Oncology, São Paulo University Medical School, Sao Paulo, Brazil

* kadantas@usp.br

## Abstract

### Background

Cryptococcosis is a devastating opportunistic infection in immunocompromised individuals, primarily in people living with HIV/AIDS. This study evaluated a protocol for the early diagnosis of meningitis due to *C. neoformans*, utilizing established molecular techniques from serum and CSF samples.

### Methods

The 18S and 5.8S (rDNA-ITS) sequence-specific nested PCR assays were compared with direct India ink staining and the latex agglutination test for detection of *C. neoformans* in serum and cerebrospinal fluid (CSF) from 49 Brazilian suspected meningitis patients. Results were validated with samples obtained from 10 patients negative for cryptococcosis and HIV, and by analysis of standard *C. neoformans* strains.

### Principal findings

The 5.8S DNA-ITS PCR was more sensitive (89–100%) and specific (100%) than the 18S rDNA PCR and conventional tests (India ink staining and latex agglutination) for identification of *C. neoformans*. While the 18S PCR exhibited a sensitivity (72%) similar to that of the latex agglutination assay in serum samples, it was superior to the latex agglutination assay when testing CSF, with a sensitivity of 84%. However, the latex agglutination was superior to the 18SrDNA PCR in specificity in CSF (92%). The 5.8S DNA-ITS PCR yielded the

**Data Availability Statement:** All data are in the manuscript and/or supporting information files.

**Funding:** The authors received no specific funding for this work.

**Competing interests:** All authors declare that they have no competing interests.

highest levels of accuracy (96–100%) of any test for detection (serological and mycological) of *C. neoformans* in both serum and CSF.

## Conclusion

Use of the nested 5.8S PCR was superior to other techniques for the diagnosis of cryptococcosis. The possibility of using serum, a non-invasively collected material, in a targeted 5.8S PCR analysis to identify Cryptococcus spp. is recommended, especially in immunosuppressed patients. Our results indicate that nested 5.8S PCR can increase the diagnostic capability of cryptococcosis, and we suggest its use to monitor patients in the future.

## Author summary

Cryptococcal meningitis is an infectious disease of global importance with high morbidity and mortality, especially among individuals with HIV/AIDS. While there have been improvements in the last two decades in the diagnosis of *Cryptococcus neoformans*, the methods presently employed are problematic for public hospitals in Brazil and other locations due to their extreme cost. In this study, we present a low-cost option for detection and identification of *C. neoformans* in noninvasive serum sample in immunosuppressed individuals, including those with HIV/AIDS. A nested PCR (5.8SrDNA-ITS) associated with the latex agglutination test has high precision in detection of suspected *Cryptococcus* spp.

## Introduction

The incidence and severity of fungal infections have increased due, in part, to the sequela of viral infections along with an elevated use of immunosuppressive therapies [1]. About 70% of individuals with HIV/AIDS develop neurological disease, and this is the initial manifestation of this infection in 10–20% of cases. Despite the gradual decline in HIV/AIDS cases, neurological manifestations of this infection remain common and are a significant cause of morbidity and mortality [2,3].

Fungi are a major contributor to opportunistic infections in those with advanced HIV infection. *Cryptococcus neoformans*, the main cause of cryptococcosis, is the most common cause of central nervous system (CNS) infection in AIDS patients [4–9]. Cryptococcal meningitis (CM) is an AIDS defining illness and the most common cause of meningitis in adults living with HIV in sub-Saharan Africa [10]. In 2014, approximately 223,100 individuals developed CM worldwide, resulting in 181,100 deaths, which represents 15% of all AIDS-related deaths [7]. In 2022, Rajasingham, et al. estimated that there are 152,000 cases (111,000–185,000) of cryptococcal meningitis, resulting in 112,000 CM-related deaths (79,000–134,000) [11]. In 2014 there was a reduction in the estimated global burden of HIV-associated Cryptococcal meningitis, probably due to the expansion of antiretroviral therapy. It has been estimated that cryptococcosis accounts for 19% of AIDS-related deaths [11]. Cases of Cryptococcal meningitis have also been reported in South and Southeast Asia. Although data are scarce for Latin America, this geographic area has the third highest incidence of *Cryptococcus* spp. in AIDS patients [12]. The persistent burden of infection suggests that death from Cryptococcal infection remains a marker for failure in the cascade of care for HIV.

In Brazil, studies on neurological diseases in HIV/AIDS patients showed Cryptococcosis to be the second leading cause of death, with a mortality rate of 45–65% (toxoplasmosis was the first and tuberculosis was the third most frequent) [13–15]. In patients with AIDS, pneumocystosis, Cryptococcosis and histoplasmosis are frequent invasive fungal infections (IFIs). They account for a high incidence of morbidity and mortality in immunocompromised patients [13–15]. In cases of neurological infections, *Cryptococcus* spp. has been reported in the south and southeast regions to have an incidence of 36 cases per 100 hospitalizations in HIV-AIDS patients/year [13–16]. The examination of cases from regional centers or from the STD/HIV-AIDS Department of the Ministry of Health form the foundation for the diagnosis of these infections [17]. The clinical signs and symptoms of Cryptococcal meningitis are indistinguishable from those of many other causes of meningitis, especially tubercular meningitis. The early detection of a Cryptococcal infection is required for optimal treatment and its identification should be given top priority to lower the unacceptable high death rate of this neglected mycosis in Brazil.

The diagnosis of *Cryptococcus* spp. is complex and typically involves a combination of symptoms, including headache, fever, cranial neuropathies, mental alteration, lethargy, memory loss and signs of meningeal irritation, confirmed by laboratory testing [17]. Currently, the laboratory diagnosis of *Cryptococcus* spp. is based on mycological, histopathological and serological tests [17–26]. The Brazilian Unified Health System (SUS) provides different methods for the diagnosis of CM. The conventional method for patients with HIV-associated infections is the direct examination of slides stained with India ink and culture of the sample in Sabouraud dextrose agar (SDA). Culture is the gold standard for the diagnosis of *Cryptococcus* spp. [21–25]. According to Vidal et al. (2018), detection of CrAg LFA in serum, CSF, and whole blood shows great sensitivity and specificity. In this retrospective review, we explored the value of other methodologies, since during the study period, CrAg LFA was not available for analysis [26].

Polymerase chain reaction (PCR)-based methods for detection of *Cryptococcus* -specific DNA or RNA sequences have the advantage of speed as they do not require culture and can be performed directly on biological samples as well as on paraffin-embedded tissues [27–37]. Specifically, detection of conserved DNA regions, including the 28S, 18S, and 5.8S ribosomal (r) DNA loci, as well as internal transcribed sites (internal transcribed spacers: ITS1-ITS4), comprise a promising and reliable alternative means of diagnosis. Indeed, in previous studies, nested PCR-based amplification of these DNA regions utilizing CSF [32–34], tissue [35], bronchoalveolar lavage fluid [36] and other samples exhibited good accuracy. In 2015, Martins et al suggested that the CN4/CN5 primer set amplified gene sequences encoding rDNA and was highly sensitive for the identification of *C. neoformans*/*C. gattii* in CSF from patients with clinical suspicion of cryptococcal meningitis [34]. However, their utilization with serum samples was associated with high rates of false-negatives and false-positives [27–36].

Drawbacks remain to conventional and molecular methods currently employed for the diagnosis of cryptococcosis. In the present study, we performed a comparative analysis of molecular (18S and 5.8S rDNA-ITS nested PCR) and conventional methods for the diagnosis of cryptococcosis. Our objective was to identify the ideal strategy for the early detection of this disease in samples of serum and CSF, minimizing the diagnostic time, hospital costs and, mainly, reducing the risk of infection progression in the patient without diagnosis.

## Methods

### Ethics statement

The study was approved by the institutional ethics committees at HCFMUSP—no. 0635/2007 and the IIER—no. 63/2007. All patients, or their responsible next of kin, provided written informed consent.

## Patients and samples

Serum and CSF were collected from 49 patients, a total of 98 samples, between January 2008 and December 2009 following their admission to the emergency units at Instituto de Infectologia Emílio Ribas (IIER) and Hospital das Clinicas da Faculdade de Medicina da Universidade de Sao Paulo (HCFMUSP) with suspected meningitis. Patients who were pregnant or younger than 18 years old were excluded.

CSF samples were analyzed for a mycological diagnosis by culture on Sabouraud-glucose medium (SAB) and direct microscopy (direct India Ink staining). After initial isolation in culture subsequent organism identification was by physiological and biochemical tests (Niger seed agar plates; Christensen's urea medium and subtyped using L-canavanine glycine bromothymol blue (CGB) medium) [37–39]. Serum samples were analysed by the IMMY *Cryptococcus* Antigen Latex Agglutination kit and were classified as HIV-positive and uninfected with cryptococcosis (group I, n = 10), HIV-negative and infected with cryptococcosis (group II, n = 7), HIV-positive and infected with other pathogens (group III, n = 14), including histoplasmosis meningitis (n = 3), paracoccididomycosis meningitis (n = 3), candidiasis meningitis (n = 2), viral meningitis (CMV, n = 2), and bacterial meningitis (*Mycobacterium tuberculosis*, n = 4); and AIDS/HIV- positive and positive for cryptococcosis (group IV; n = 18). In this last group the patients had $CD4^+$ T lymphocyte counts less than 200 cells/mL. In the absence of direct pathogen isolation from patient samples, diagnosis was confirmed by histopathology or, in some cases, at autopsy. All samples underwent serological testing at both institutions for HIV-1/2, hepatitis B (HBsAg and anti-HBc), antibodies to hepatitis C, VDRL test for syphilis and for fungal pathogens (*Paracoccidioides brasiliensis*, *Histoplasma capsulatum*, *Candida* spp., *Aspergillus fumigatus*, and *Cryptococcus* spp.).

## Control strains

To assess the analytical sensitivity of the qualitative PCR assays the initial number of *Cryptococcus* spp. cells present ($10^6$/mL) was determined by microscopic examination (Neubauer camera). A mixed CSF and serum aliquot was spiked with $10^6$ fungi cells/mL. After homogenization, this sample was used to prepare 10 ten-fold serial dilutions ranging from $10^6$ to 1 cell of *C. neoformans*/mL (ATCC 24067 and RI 02) and *C. gattii*/mL (TIPI01). DNA extraction was performed on each of the spiked samples. All assays were performed in triplicate.

The specificity of molecular methods was evaluated using isolates well characterized by the ATCC and CDC. They consisted of paracoccidiodomycosis (*P. brasiliensis*—18 and B-339 ATCC 32069), nocardiosis (*Nocardia* spp.), candidiasis (*Candida albicans* and *C. parapsilosis*), histoplasmosis [*H. capsulatum*, ATCC 28308 (CDC: B973), ATCC 12700 (CDC: A811), and HC200 (GenBank: DQ239887)] and aspergillosis (*Aspergillus* spp.).

Analysis of the 18S and 5.8S (rDNA-ITS) sequence-specific nested PCRs for the detection of *C. neoformans* and *C. gattii* in serum, CSF and cell suspensions of culture isolates yielded a sensitivity of 1 cell/mL for CSF and 10 cells/mL for serum. All assays were performed in triplicate and showed 100% reproducibility. In addition, the nested PCRs with 18SrDNA and 5.8SrDNA-ITS were highly specific for *Cryptococcus* spp. and had no cross-reactivity with other pathogens.

## Mycology

**Processing of samples.** A standard method was followed for the processing of samples [37]. CSF was centrifuged at 1000 g for 15 min (Eppendorf 5810R, Hamburg—Germany). The pellet was used for investigation by culture, direct microscopy (India ink wet mount, Gram stain) and PCR, while the supernatant was tested by serology (latex agglutination analysis) and

molecular (18S and 5.8SrDNA) analysis. All samples were tested undiluted. Clotted blood was centrifuged at 1000 rpm. for 5 min to obtain serum [37] and subjected to serological (latex agglutination) and molecular (18S and 5.8SrDNA) analyses.

CSF was cultivated on SAB agar [Difco Laboratories, Detroit, MI] at 25˚C and the isolates obtained were analyzed by physiological and biochemical tests, consisting in subculture on Niger seed agar plates (Guizotia abyssinica), as described by Kwon-Chung and Bennett following incubation at 37˚C for two weeks [8] and, for urease detection using Christensen's urea medium (Becton Dickinson, New Jersey, USA) [8] and subtyped using L-canavanine glycine bromothymol blue (CGB) medium (Difco Laboratories, Detroit, MI) [37,38]. After identification, the isolates were stored in 80% glycerol (Sigma-Aldrich, St. Louis, MO, USA) at -80˚C.

## Serology

Serum and CSF samples were subjected to latex agglutination analysis to detect *C. neoformans* antigens, using the IMMY *Cryptococcus* Antigen Latex Agglutination kit (Immuno-Mycologics, Inc, Norman, OK, USA), according to the manufacturer's instructions [40].

## DNA extraction

To extract DNA from cell cultures, CSF and sera, 200 μL samples were admixed with 40 μL of 60 mg/mL lysing enzyme from *Thichoderma harzium* (cat. no. L1412, Sigma Chemical Co., St. Louis, MO, USA) in 1 M sorbitol, 100 mM EDTA, and 14 mM β-mercaptoethanol. Samples were then incubated for 30 min at 30˚C and centrifuged at 5,000 ×g (Eppendorf, Hamburg, Germany) at room temperature. Precipitated cells were resuspended in 180 μL ATL buffer (QIAamp DNA Mini Kit, Qiagen, Hilden, Germany), and lysed for 3 h at 56˚C with 100 mg/mL proteinase K. DNA was then extracted using the QIAamp DNA Mini Kit for cell and QIAamp Blood DNA Mini Kit (Qiagen) with CSF and serum samples.

## Nested PCR

A nested PCR was utilized to detect 5.8S and 18S *C. neoformans* rDNA sequences in the biological and culture samples. Table 1 describes the outer primers designed to amplify regions of the 5.8S (600-base pair [bp] product) and 18S (429-bp product) sequences that are highly conserved in several human fungal pathogens, as well as the inner primers designed to amplify regions of the 5.8S (116-bp) and 18S (278-bp) sequences that are specific to *Cryptococcus* spp. and to *C. neoformans* [30,35,41]. Prior to the *Cryptococcus* spp. assay, patient samples were subjected to nested PCR for the housekeeping gene human *glyceraldehydes-3-phosphate dehydrogenase* (*GADPH*; GenBank accession number J04038.1) [42], using primers described in Table 1. GADPH, was confirmed in all samples the presence of human DNA and the absence of PCR inhibitors. All assays were performed in triplicate and showed 100% reproducibility.

## Multiplex PCR: *Cryptococcus* spp

For differentiation of distinct *Cryptococcus* spp., a multiplex PCR was performed using two pairs of primers designed to amplify a 564-bp and a 448-bp product specific for *C. neoformans* and for *C. gattii* [29] (Table 1). All amplification experiments were performed in triplicate and included both a negative (no added DNA) and a positive control. To avoid contamination of the PCR mixtures, the preparation and addition of the template DNA were performed in separate rooms.

**Table 1. Description of the primer's sequences.**

| PCR | | | Primers 5'→3' | | Fragment [bp] | References |
|---|---|---|---|---|---|---|
| Nested PCR | 18S rDNA | Fungus I | GTT AAA AAG CTC GTA GTT G | | 429 | Bialek et al. (2002) [35] |
| | | Fungus II | TCC CTA GTC GGC ATA GTT TA | | | |
| | | Crypto I | TCC TCA CGG AGT GCA CTG TCT TG3' | | 278 | |
| | | Crypto II | CAG TTG TTG GTC TTC CGT CAA TCT A | | | |
| | 5,8 S rDNA-ITS | ITS1 | TCC GTA GGT GAA CCT GCG G | | ITS1-ITS4-600-650 | Mitchell et al. (1994) [30] Fugita et al., (2001) [41] |
| | | ITS4 | TCC TCC GCT TAT TGA TAT GC | | | |
| | | ITS3 | GCATCGATGAAGAACGCAGC | | ITS3-ITS4-264-432 | |
| | | CN5 | GAA GGG CAT GCC TGT TTG AGA G | | 116 | |
| | | CN6 | TTT AAG GCG AGC CGA CGT CCT T | | | |
| | GAPDH | gapI | GAC AAC AGC CTC AAG ATC ATC | | 610 | Escolani et al. (1988) [42] |
| | | gapII | GAC GGC AGG TCA GGT CCA CCA | | | |
| | | gapIII | AAT GCC TCC TGC ACC ACC | | 248 | |
| | | gapIV | ATG CCA GTG AGC TTC CCG | | | |
| Multiplex PCR | C. neoformans | MX1cn | ATT GCG TCC ACC AAG GAG CTC | | 564 | Casali et al. (2003) [29] |
| | | MX2cn | ATT GCG TCC ATG TTA CGT GGC | | | |
| | C. gattii | MX3cg | ATT GCG TCC AAG GTG TTT GTT G | | 448 | |
| | | MX4cg | ATT GCG TCC ATC AAC CG TTA TC | | | |

A nested PCR was utilized to detect the 5.8SrDNA-ITS and 18S rDNA. The following outer primers were designed to amplify regions of the 18S (429 bases pair [bp] product) and 5.8S (600-base pair [bp] product) sequences highly conserved in several human fungal pathogens. Conversely, the following inner primers were designed to amplify regions of the 18S (278-bp) and 5.8S (116-bp) sequences specific to *Cryptococcus* spp. and *C. neoformans*.

Nested GADPH—glyceraldehydes-3-phosphate dehydrogenase

Multiplex PCR MXcn and MX2cn–a multiplex PCR was performed using two pairs of primers designed to amplify a 564-bp product specific to *C. neoformans*;

Multiplex PCR MX3cg and MX4cg—multiplex PCR was performed using two pairs of primers designed to amplify a 448-bp product specific to *C. gattii*

bp—base pair

## Sequencing analysis

All PCR reagents were obtained from Invitrogen (Carlsbad, CA, USA), and samples were processed and amplified three times on a Veriti 96 thermocycler (Applied Biosystems, Life Technologies Corporation, Carlsbad, CA, USA). All assays included negative controls without DNA and positive controls with DNA from *H. capsulatum* ATCC A811 and B923, *C. neoformans* ATCC 24067, and *P. brasiliensis* 18 and 339. Products were electrophoresed on 1.5% agarose, stained with ethidium bromide, and visualized on a UV transilluminator.

The amplification products obtained from the 5.8S rDNA-ITS region were purified using a PureLink kit [Invitrogen (Life Technologies), Waltham, MA, USA] and sequenced using a MegaBACE 1000 DNA Sequencing System (GE Healthcare, Little Chalfont, United Kingdom) and a DYEnamic ET Dye Terminator Kit (with Thermo Sequence II DNA Polymerase, GE Healthcare), according to the manufacturer's protocol. All sequences were analyzed with Sequence Analyzer software using the Base Caller Cimarron 3.12 program and were identified by BLAST analysis (http://www.ncbi.nlm.nih.gov/BLAST). Multiple alignments were assembled using Clustal W software (http://www.ebi.ac.uk/clustaw).

## Statistical analysis

The comparison between independent groups of patients, with a positive or negative diagnosis of Cryptococcosis, were performed using the Student t test for continuous variables and either the chi-square test or Fisher's exact test for categorical variables (gender and clinical variables). Agreement between the diagnostic methods was assessed by inter-rater agreement [43]

**Table 2. Kappa Index Interpretation [43].**

| Kappa statistic | Strength of agreement |
|---|---|
| < 0.00 | Poor |
| 0.00–0.20 | Slight |
| 0.21–0.40 | Fair |
| 0.41–0.60 | Moderate |
| 0.61–0.80 | Substantial |
| 0.81–1.00 | Almost perfect |

(Cohen's Kappa et al.,1960) as interpreted using the Landis and Koch-Kappa Benchmark Scale, in Table 2 [44]. Statistical analyses were performed using the software package SPSS (v. 24.0, IBM, New York, NY, USA). The significance level was set at p ≤ 0.05. The sensitivity, specificity, positive and negative predictive values, and accuracy were analyzed according to Fletcher et al 1996 [45].

## Results

Patients with or without cryptococcal infection were similar in demographic variables: gender, age and ethnicity (Table 3). The most frequent underlying disease was HIV/AIDS (n = 32; 65.3%), of which 78.3% were also positive for *Cryptococcus* spp. The other frequent underlying diseases were malignancy, diabetes, leukemia, and Hodgkin's disease (Table 3). Cryptococcal meningitis was diagnosed in four HIV-negative patients while 10 cases had meningitis due to other pathogens. In three of the HIV-negative patients their disease progressed to disseminated Cryptococcal disease. Twenty percent of the cases that progressed to death were HIV-positive and presented with disseminated Cryptococcosis. Only two patients who were negative for *Cryptococcus* spp. died, one associated with CMV and the other with histoplasmosis.

Table 4 provides information on the reliability of the various tests for the detection of cryptococcosis. The direct India ink test did not detect *Cryptococcus* spp. in sera from both HIV-positive and HIV-negative subjects. However, this test showed a substantial agreement with the culture gold standard when analyzing CSF. The kappa index for the India ink test was similar in HIV-positive (K = 0.695 (0.576–0.814); p<0.001) and HIV-negative (K = 0.611 (0.423–0.799); p = 0.006) samples. The results of the multiplex PCR and sequencing showed greater efficacy than using L-canavanine glycine bromothymol blue (CGB) medium in differentiating between species. This confirmed that 77.4% of the isolates were *C. neoformans* while 22.3% were inconclusive. However, the multiplex PCR test confirmed that 100% of the isolates were *C. neoformans*. Sequencing showed that all samples were *C. neoformans* with 98% similarity.

The latex agglutination test detected *Cryptococcus* spp. in both sera and CSF (Table 4). Overall, the latex agglutination test had a higher sensitivity when assaying sera (72%) as compared to CSF (56%). When applied to CSF this test exhibited only a moderate accuracy, with a similar low sensitivity and negative predictive value (NPV), in both HIV-positive and -negative subjects. However, differences between HIV-positive and HIV-negative patients were seen in the serum analyses. HIV-negative patients' sera had a substantial level of accuracy while in the HIV-positive subjects' sera accuracy was only moderate. In HIV-negative patients this test had 100% specificity for both sera and CSF.

The latex agglutination test in the seropositive patients yielded four false-positive tests for *Cryptococcus* spp., subsequently identified by molecular methods as *P. brasiliensis*, *H. capsulatum* and bacterial meningitis.

**Table 3. Demographic and clinical data of study subjects.**

| Variable | Cryptococcosis | | Statistical Analysis |
|---|---|---|---|
| | Positive n = 25 | Negative n = 24 | |
| **Age (years)** | | | **t test** |
| Mean ± SD | 38.6±10.7 | 39.3±11.0 | $t_{(47)} = 0.210$; p = 0.835 |
| **Gender n = 49 (%)** | | | **chi-square test** |
| Male n = 33 (67.4%) | 19 (76.0%) | 14 (58.3%) | $X^2_{(1)} = 1.738$; p = 0.187 |
| Female n = 16 (32.6%) | 6 (24.0%) | 10 (41.7%) | |
| **Ethnicity n = 49 (%)** | | | **Fisher's exact test** |
| White n = 34 (69.4%) | 18 (72.0%) | 16 (66.7%) | $X^2_{(2)} = 0.352$; p = 0.898 |
| Mixed race n = 11 (22.4%) | 5 (20.0%) | 6 (25.0%) | |
| Black n = 4 (8.2%) | 2 (8.0%) | 2 (8.3%) | |
| **Underling diseases n = 49 (%)** | | | **Fisher's Exact Test** |
| HIV/AIDS | 18 (78.3%) | 14 (58.3%) | $X^2_{(4)} = 4.124$; p = 0.494 |
| Malignancy | 2 (8.7%) | 6 (25.0%) | |
| Diabetes | 1 (4.4%) | 2 (8.3%) | |
| Leukemia | 1 (4.4%) | 2 (8.3%) | |
| Hodgkin's Disease | 1 (4.4%) | 0 (0.0%) | |
| Missing | 2 | 0 | |
| **HIV n = 49 (%)** | | | **chi-square test** |
| Positive n = 32 (65.3%) | 18 (72.0%) | 14 (58.3%) | $X^2_{(1)} = 1.009$; p = 0.315 |
| Negative n = 17 (34.7%) | 7 (28.0%) | 10 (41.7%) | |
| **Underling diseases n = 49 (%)** | | | **Fisher's Exact Test** |
| **Outcome n = 49 (%)** | | | **Fisher's exact Test** |
| Alive | 20 (80.0%) | 22 (91.7%) | $X^2_{(1)} = 1.403$; p = 0.226 |
| Dead | 5 (20.0%) | 2 (8.3%) | |
| **Clinical Forms n = 49 (%)** | | | **Fisher's exact Test** |
| HIV + n = 32 | | | |
| Disseminated n = 18 (56.3%) | 18 (100.0%) | 0 (0.0%) | |
| Meningitis n = 14 (43.8%) | 0 (0.0%) | 14 (100.0%) | |
| HIV—n = 17 | | | $X^2_{(1)} = 5.204$; p = 0.051 |
| Disseminated n = 3 (17.6%) | 3 (42.9%) | 0 (0.0%) | |
| Meningitis n = 14 (82.4%) | 4 (57.1%) | 10 (100.0%) | |
| **Meningitis (another agents) n = 24 (%)** | | | |
| Bacteria n = 9 (34.6%) 4 Patients HIV + | | | |
| *Mycobacterium tuberculosis* | 0 (0.0%) | 3 (33.3%) | |
| Missing | 0 (0.0%) | 6 (66.7%) | |
| Fungal n = 8 (30.8%) 8 Patients HIV + | | | |
| *Histoplasma capsulatun* | 0 (0.0%) | 3 (37.5%) | |
| *Paracoccidioides brasiliensis* | 0 (0.0%) | 3 (37.5%) | |
| *Candida spp.* | 0 (0.0%) | 2 (25.0%) | |
| Virus n = 7 (26.9%) 2 patients HIV + | | | |
| Cytomegalovirus | 0 (0.0%) | 2 (28.6%) | |
| Missing | 0 (0.0%) | 5 (71.4%) | |
| Protozoan n = 2 (7.7) 2 patients HIV - | | | |
| *Toxoplasma gondii* | 2 (100.0%) | 0 (0.0%) | |

The laboratories from the Institute of Tropical Medicine, IIER, and HCFMUSP, carried out the identification of fungi, bacteria, and viruses, according to the Ministry of Health's guidelines.

(**https://bvsms.saude.gov.br/bvs/publicacoes/guia_vigilancia_saude_volume_1.pdf**)

**Table 4. Accuracy of the diagnostic techniques for the diagnosis of Cryptococcosis.**

| Diagnostic Method | Clinical Sample | | Gold Standard % (N) | | TOTAL | Analysis | | | | | |
|---|---|---|---|---|---|---|---|---|---|---|---|
| | | | + | - | | Sensitivity | Specificity | PPV | NPV | Accuracy | Kappa index (CI 95%); p |
| **Indian Ink** | CSF: HIV+ | + | 40.6 (13) | 0.0 (0) | 40.6 (13) | 0.72 | 1.00 | 1.00 | 0.74 | 0.84 | K = 0.695 (0.576–0.814); p<0.001 |
| | | - | 15.6 (5) | 43.8 (14) | 59.4 (19) | | | | | | |
| | TOTAL | | 56.3 (18) | 43.8 (14) | 100.0 (32) | | | | | | |
| | CSF: HIV - | + | 23.5 (4) | 0.0 (0) | 23.5 (4) | 0.57 | 1.00 | 1.00 | 0.77 | 0.82 | K = 0.611 (0.423–0.799); p = 0.006 |
| | | - | 17.6 (3) | 58.8 (10) | 76.5 (13) | | | | | | |
| | TOTAL | | 41.2 (7) | 58.8 (10) | 100.0 (17) | | | | | | |
| | CSF: Total | + | 34.7 (17) | 0.0 (0) | 34.7 (17) | 0.68 | 1.00 | 1.00 | 0.75 | 0.84 | K = 0.675 (0.576–0.774); p<0.001 |
| | | - | 16.3 (8) | 49.0 (24) | 65.3 (32) | | | | | | |
| | TOTAL | | 51.0 (25) | 49.0 (24) | 100.0 (49) | | | | | | |
| **Latex agglutination analysis** | Serum: HIV+ | + | 40.6 (13) | 6.3 (2) | 46.9 (15) | 0.72 | 0.86 | 0.87 | 0.71 | 0.78 | K = 0.566 (0.423–0.709); p = 0.001 |
| | | - | 15.6 (5) | 37.5 (12) | 53.1 (17) | | | | | | |
| | TOTAL | | 56.3 (18) | 43.8 (14) | 100.0 (32) | | | | | | |
| | Serum: HIV- | + | 29.4 (5) | 0.0 (0) | 29.4 (5) | 0.71 | 1.00 | 1.00 | 0.83 | 0.88 | K = 0.746 (0.583–0.909); p = 0.001 |
| | | - | 11.8 (2) | 58.8 (10) | 70.6 (12) | | | | | | |
| | TOTAL | | 41.2 (7) | 58.8 (10) | 100.0 (17) | | | | | | |
| | Serum: Total | + | 36.7 (18) | 4.1 (2) | 40.8 (20) | 0.72 | 0.92 | 0.9 | 0.76 | 0.82 | K = 0.634 (0.536–0.742), p<0.001 |
| | | - | 14.3 (7) | 44.9 (22) | 59.2 (29) | | | | | | |
| | TOTAL | | 51.0 (25) | 49.0 (24) | 100.0 (49) | | | | | | |
| | CSF: HIV+ | + | 34.4 (11) | 6.3 (2) | 40.6 (13) | 0.61 | 0.86 | 0.85 | 0.63 | 0.72 | K = 0.450 (0.302–0.598); p = 0.007 |
| | | - | 21.9 (7) | 37.5 (12) | 59.4 (19) | | | | | | |
| | TOTAL | | 56.3 (18) | 43.8 (14) | 100.0 (32) | | | | | | |
| | CSF: HIV- | + | 17.6 (3) | 0.0 (0) | 17.6 (3) | 0.43 | 1.00 | 1.00 | 0.71 | 0.76 | K = 0.469 (0.272–0.666), p = 0.023 |
| | | - | 34.7 (4) | 58.8 (10) | 82.4 (14) | | | | | | |
| | TOTAL | | 41.2 (7) | 58.8 (10) | 100.0 (17) | | | | | | |
| | CSF: Total | + | 28.6 (14) | 4.1 (2) | 32.7 (16) | 0.56 | 0.92 | 0.88 | 0.67 | 0.73 | K = 0.473 (0.356–0.590); p<0.001 |
| | | - | 22.4 (11) | 44.9 (22) | 67.3 (33) | | | | | | |
| | TOTAL | | 51.0 (25) | 49.0 (24) | 100.0 (49) | | | | | | |
| **PCR 18S** | Serum: HIV+ | + | 40.6 (13) | 6.3 (2) | 46.9 (15) | 0.72 | 0.86 | 0.87 | 0.71 | 0.78 | K = 0.566 (0.423–0.709); p = 0.001 |
| | | - | 15.6 (5) | 37.5 (12) | 53.1 (17) | | | | | | |
| | TOTAL | | 56.3 (18) | 43.8 (14) | 100.0 (32) | | | | | | |
| | Serum: HIV- | + | 29.4 (5) | 0.0 (0) | 29.4 (5) | 0.71 | 1.00 | 1.00 | 0.83 | 0.88 | K = 0.746 (0.583–0.909); p = 0.001 |
| | | - | 11.8 (2) | 58.8 (10) | 70.6 (12) | | | | | | |
| | TOTAL | | 41.2 (7) | 58.8 (10) | 100.0 (17) | | | | | | |
| | Serum: Total | + | 36.7 (18) | 4.1 (2) | 40.8 (20) | | | | | | |
| | | - | 14.3 (7) | 44.9 (22) | 59.2 (29) | 0.72 | 0.92 | 0.9 | 0.76 | 0.82 | K = 0.634 (0.526–0.742); p<0.001 |
| | TOTAL | | 51.0 (25) | 49.0 (24) | 100.0 (49) | | | | | | |
| | CSF: HIV+ | + | 46.9 (15) | 12.5 (4) | 59.4 (19) | 0.83 | 0.71 | 0.79 | 0.77 | 0.78 | K = 0.552 (0.403–0.701); p = 0.002 |
| | | - | 9.4 (3) | 31.3 (10) | 40.6 (13) | | | | | | |
| | TOTAL | | 56.3 (18) | 43.8 (14) | 100.0 (32) | | | | | | |
| | CSF: HIV- | + | 35.3 (6) | 0.0 (0) | 35.3 (6) | 0.86 | 1.00 | 1,00 | 0.91 | 0.94 | K = 0.876 (0.757–0.995); p<0.001 |
| | | - | 5.9 (1) | 58.8 (10) | 64.7 (11) | | | | | | |
| | TOTAL | | 41.2 (7) | 58.8 (10) | 100.0 (17) | | | | | | |
| | CSF: Total | + | 42.9 (21) | 8.2 (4) | 51.0 (25) | 0.84 | 0.83 | 0.84 | 0.83 | 0.84 | K = 0.673 (0.567–0.779); p<0.001 |
| | | - | 8.2 (4) | 40.8 (40) | 49.0 (24) | | | | | | |
| | TOTAL | | 51.0 (25) | 49.0 (24) | 100.0 (49) | | | | | | |

*(Continued)*

**Table 4.** (Continued)

| Diagnostic Method | Clinical Sample | | Gold Standard % (N) | | TOTAL | Analysis | | | | | |
|---|---|---|---|---|---|---|---|---|---|---|---|
| | | | + | - | | Sensitivity | Specificity | PPV | NPV | Accuracy | Kappa index (CI 95%); p |
| **PCR 5.8S** | Serum: HIV+ | + | 50.0 (16) | 0.0 (0) | 50.0 (16) | 0.89 | 1.00 | 1.00 | 0.88 | 0.94 | K = 0.875 (0.790–0.960); p<0.001 |
| | | - | 6.3 (2) | 43.8 (14) | 50.0 (16) | | | | | | |
| | TOTAL | | 56.3 (18) | 43.8 (14) | 100.0 (32) | | | | | | |
| | Serum: HIV- | + | 41.2 (7) | 0.0 (0) | 41.2 (7) | 1.00 | 1.00 | 1.00 | 1.00 | 1.00 | K = 1; p<0.001 |
| | | - | 0.0 (0) | 58.8 (10) | 58.8 (10) | | | | | | |
| | TOTAL | | 41.2 (7) | 58.8 (10) | 100.0 (17) | | | | | | |
| | Serum: Total | + | 46.9 (23) | 0.0 (0) | 46.9 (23) | 0.92 | 1.00 | 1.00 | 0.92 | 0.96 | K = 0.918 (0.862–0.974); p<0.001 |
| | | - | 4.1 (2) | 49.0 (24) | 53.1 (26) | | | | | | |
| | TOTAL | | 51.0 (25) | 49.0 (24) | 100.0 (49) | | | | | | |
| | CSF: HIV+ | + | 56.3 (18) | 0.0 (0) | 56.3 (18) | 1.00 | 1.00 | 1,00 | 1.00 | 1.00 | K = 1; p<0.001 |
| | | - | 0.0 (0) | 43.8 (14) | 43.8 (14) | | | | | | |
| | TOTAL | | 56.3 (18) | 43.8 (14) | 100.0 (32) | | | | | | |
| | CSF: HIV- | + | 41.2 (7) | 0.0 (0) | 41.2 (7) | 1.00 | 1.00 | 1,00 | 1.00 | 1.00 | K = 1; p<0.001 |
| | | - | 0.0 (0) | 58.8 (10) | 58.8 (10) | | | | | | |
| | TOTAL | | 41.2 (7) | 58.8 (10) | 100.0 (17) | | | | | | |
| | CSF: Total | + | 51.0 (25) | 0.0 (0) | 51.0 (25) | 1.00 | 1.00 | 1,00 | 1.00 | 1.00 | K = 1; p<0.001 |
| | | - | 0.0 (0) | 49.0 (24) | 49.0 (24) | | | | | | |
| | TOTAL | | 51.0 (25) | 49.0 (24) | 100.0 (49) | | | | | | |

PPV, positive predictive value; NPV, negative predictive value

Strength agreement between the reference method and molecular methods was assessed by inter-rater agreement

The 18S PCR assay results in HIV-positive patients were similar for both serum (K = 0.566; 0.423–0.709; p = 0.001) and CSF (K = 0.552; 0.403–0.701; p = 0.002), and agreement was only moderate. The Kappa index for the HIV-negative patients was 0.746; (0.583–0.909; p = 0.001) for serum and 0.876 (0.757–0.995; p<0.001) for CSF, and agreement was substantial. Comparative analysis of serum samples did not demonstrate a difference between the 18SrDNA PCR and latex agglutination assays. These results demonstrate that latex agglutination can detect early stage *Cryptococcus* spp. In CSF samples from HIV-positive patients, the 18S PCR demonstrated a better performance in terms of sensitivity. The latex agglutination test was the most specific, but the 18S PCR had a high sensitivity, and can be used as a complementary test. The CSF samples showed a higher accuracy (0.94) and a kappa index (0.876; 0.757–0.995; p<0.001) in the HIV-negative patients with a high level of agreement with the culture gold standard.

The 5.8S PCR assay demonstrated a high degree of accuracy in CSF samples from both HIV-positive and -negative patients. Analysis of the corresponding serum samples revealed a comparatively lower sensitivity and NVP, despite both having an almost perfect agreement (Kappa index 1; p<0.001 for CSF and 0.918 (0.862–0.974); p<0.001 for serum).

In order to select a highly accurate methodology for the diagnosis of Cryptococcosis, we compared the results obtained by analysis of the 18S and 5.8S PCR-generated DNA sequences with results of the latex agglutination test and India ink staining. The 5.8S primers showed 100% specificity for detection of *Cryptococcus* sp. Comparing the India ink results with the 18S and 5.8S PCR assays, the genomic methods were the most sensitive for detection of *Cryptococcus* spp. both in CSF and serum. Regarding specificity, no difference was observed between the analysis using India ink and the 5.8S PCR primers. In contrast, the 18S PCR showed a lower specificity compared to India ink staining.

When we compared the latex agglutination test with the 5.8S PCR, the PCR had a higher sensitivity and specificity for detection of *Cryptococcus* spp. in both serum and CSF from HIV-positive and -negative individuals. However, the 18S PCR was less specific than the latex agglutination test when assaying CSF samples from the HIV-positive patients.

The comparative analysis between conventional (mycological and serological) and molecular (18S and 5.8S PCR) methods showed that the molecular methods had a higher sensitivity and specificity (Table 4). The 5.8S PCR exhibited the highest sensitivity and specificity with an accuracy of 100%; the Kappa index analysis showed almost perfect agreement with the gold standard (culture) in serum samples HIV-positive (0.940; K = 0.875; 0.790–0.960; p<0.001) and HIV-negative (1.000; K = 1; p<0.001) and CSF (1.000; K = 1; p<0.001—HIV-positive and 1.000 K = 1; p<0.001—HIV-negative), in both groups. Two serum samples collected from HIV-positive patients with a positive culture for Cryptococcosis were 5.8S PCR-negative. These two samples were also negative by the 18S PCR and latex agglutination test, and the corresponding CSF samples were also negative in all tests. Lastly, there were no significant differences between serum and CSF utilizing any of the analytical methods on patients with disseminated Cryptococcosis and *Cryptococcus* meningitis.

By comparing the clinical diagnosis among the 25 patients with Cryptococcosis, 4 had meningitis and 21 had disseminated disease. It was observed that the diagnosis did not influence the performance of the diagnostic tests.

## Discussion

Despite the observed decline of Cryptococcal meningitis in patients with HIV/AIDS, its occurrence still represents a determinant of poor prognosis in HIV-infected patients [3,5,6]. The presence of Cryptococcosis in the central nervous system leads to severe neurological complications [2,3,7,10]. In recent years, great advances have been described in development of laboratory techniques for the diagnosis of infectious diseases and in the emergence of new methodologies for the diagnosis of *Cryptococcus* spp., especially in HIV-positive individuals [21–26,28,32–36,40,41]. However, the problem of early accurate diagnosis still remains. In this study, we analyzed two sets of primer pairs, 18SrDNA and 5.8SrDNA, in a nested PCR analysis and demonstrated high accuracy in detection of *Cryptococcus* spp. in both serum and CSF. The success of early diagnosis of Cryptococcosis, especially in non- invasive serum samples, is extremely relevant. Both assays were validated for detection and identification of *Cryptococcus* spp. with high accuracy in serum samples.

In mycological and histopathological tests that are the gold standard for PCR validation, CSF cultures were more frequently positive for *C. neoformans* in HIV-positive patients than in HIV-negative patients. This was despite the reported difficulties in isolating *Cryptococcus* spp. in HIV positive patients by culture [46,47]. The low sensitivity and accuracy of the India ink test and histopathology confirmed what has been described previously [48,49]. The relatively high lower limit of detection in these two assays is a significant disadvantage as it can lead to false-negative results, especially for those tested during the early stages of their disease. False negative results combined with the nonspecific symptoms of cryptococcosis can lead to misdiagnosis, progression of cryptococcosis and death [34].

Diagnosis of Cryptococcosis in sera by the latex agglutination test yielded a higher sensitivity and NPV than did the comparable analysis of CSF [50,51]. However, both samples have been reported to show low accuracy in this test [52–54]. The findings suggest the absence or reduced burden of *Cryptococcus* spp in current. In addition, false-positive results in sera and CSF from HIV- negative patients can be due to cross-reactions with other opportunistic pathogens, such as *Trichosporon beigelii*, *Klebsiella pneumonia*, *Capnocytophaga canimorsus* or *Stomatococcus mucilaginosus*, making diagnostic confirmation difficult [19,54].

Several studies based on molecular methods have shown that *C. neoformans* exhibits different degrees of genetic heterogeneity between clinical and environmental isolates. The primer sets used in these *Cryptococcus* studies were those normally used for genotyping characterization [5,46]. However, there are few studies evaluating the effectiveness of this methodology in molecular diagnosis. In the present study we chose two sets of primers widely used for *Cryptococcus* genotyping. The set of 18SrDNA and 5.8SrDNA primers that amplified a fragment of 278 bp and 116 bp, respectively from a specific region of the gene that encodes rDNA [5,46]

PCR results with 18SrDNA primers showed 100% specificity for direct detection of *Cryptococcus* sp. in CSF and serum samples from the HIV-negative group. However, a different profile was observed in HIV-positive patients. The sensitivity in this latter group was less than 90%. A similar finding was reported by Bialek et al. [35] with mouse brain tissue. The comparative analysis demonstrated that the 18SrDNA PCR and latex agglutination assays for early detection of *Cryptococcus* spp. in CSF samples from HIV-positive patients can be complementary. The 18S PCR exhibited better performance in terms of sensitivity [55], while the latex agglutination test was more specific. One hypothesis for the lower sensitivity and specificity of 18S primers in CSF samples may be interference in the gene amplification process by cell wall components of *Cryptococcus* spp. or even due to interference in the treatment being applied [18].

Of the analyzed primers, the 5.8S PCR showed 100% sensitivity and specificity with CSF samples, consistent with previous reports [21,31,34]. According to these studies, despite its high accuracy, the 5.8S PCR is not indicated for monitoring patients, as false-negative results were observed shortly after treatment initiation. However, our findings indicate that the nested PCR using the 5.8S primers in both CSF and serum samples was highly accurate in patients with suspected cryptococcal meningitis, regardless of the presence/absence of concomitant treatment. Despite the interspecies analysis showing optimal specificity (100%) with a high positive predictive value (1.00), the analysis of serum from HIV-positive and HIV-negative patients was almost perfect using the 5.8S PCR (K = 0.918 (0.862–0.974); p< 0.001). By comparison, the 18S-CRP had only moderate to substantial success in patients with or without HIV infection (K = 0.634 (0.526–0.742); p<0.001). Two cases from the CM/AIDS group were false negatives, possibly due to low or non-uniform distribution of specimens from which DNA was extracted. Furthermore, utilization of the 5.8S primers for the analysis of serum samples is a non-invasive and low-cost technique.

The use of a set of multiplex primers for molecular diagnosis could be an excellent alternative, since in the same reaction it is possible to detect the causative organism and, concurrently genotype the infecting strain. Multiplex PCR and sequencing data confirmed the detection of *C. neoformans* in biological samples from AIDS patients worldwide, including Brazil [56–59].

A limitation of the present study was the sample size since the number of HIV-negative patients was lower compared to the HIV-positive subjects. This reduced the power of the statistical evaluation. In addition, patients' medical records did not contain all relevant information, such as occupational risk factors. In addition, the analysis of patients' follow-up was limited to their hospitalization and so the risk of recurrence or other complications after discharge could not be assessed. Lastly, as the study was retrospective, it was not possible to consider the test lateral flow assay cryptococcal antigen (CrAg LFA) when testing these samples.

Our results indicate that the nested 5.8S PCR can increase the ability to diagnose cryptococcosis, and we suggest its use to monitor patients in the future. Furthermore, the possibility of using serum, a non-invasively collected material, in a targeted 5.8S PCR analysis to identify *Cryptococcus* spp. is worth highlighting, especially in immunosuppressed patients.

Patients' sera were evaluated by both serological and molecular tests, since the purpose of our study was to identify a test that could be used on non-invasively collected samples, mainly

from immunosuppressed patients, for the early diagnosis of *Cryptococcus* infection. The nested PCR specific for the 5.8Sr DNA sequence was shown to be superior to both conventional detection methods and 18S PCR for *Cryptococcus* detection.

## Acknowledgments

We are grateful to Dr. Shigueko Sonohara Troyano Pueyo, Dr. Claudio Mendes Pannuti and Cidia Vasconcelos for their scientific support during the development of this research.

## Author Contributions

**Conceptualization:** Katia Cristina Dantas.

**Formal analysis:** Vera Lúcia Teixeira de Freitas.

**Investigation:** Katia Cristina Dantas, Alfredo Mendroni Júnior, Marcos Vinicius da Silva, Paulo Ricardo Criado.

**Methodology:** Katia Cristina Dantas, Roseli Santos de Freitas—Xavier, Suzete Cleusa Ferreira Spina Lombardi.

**Project administration:** Katia Cristina Dantas.

**Resources:** Terezinha Morato Bastos de Almeida.

**Supervision:** Katia Cristina Dantas.

**Validation:** Vera Lúcia Teixeira de Freitas.

**Writing – original draft:** Katia Cristina Dantas, Roseli Santos de Freitas—Xavier, Suzete Cleusa Ferreira Spina Lombardi, Vera Lúcia Teixeira de Freitas.

**Writing – review & editing:** Katia Cristina Dantas, Roseli Santos de Freitas—Xavier, Vera Lúcia Teixeira de Freitas, Terezinha Morato Bastos de Almeida.

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
