## [Decision Letter · Decision Letter 0]

17 Nov 2022

Dear Dr Dantas,

Thank you very much for submitting your manuscript "Comparative analysis of diagnostic methods to detection Cryptococcus neoformans meningitis" for consideration at PLOS Neglected Tropical Diseases. As with all papers reviewed by the journal, your manuscript was reviewed by members of the editorial board and by several independent reviewers. In light of the reviews (below this email), we would like to invite the resubmission of a significantly-revised version that takes into account the reviewers' comments. 

We cannot make any decision about publication until we have seen the revised manuscript and your response to the reviewers' comments. Your revised manuscript is also likely to be sent to reviewers for further evaluation.

Sincerely,

Marcio L Rodrigues

Section Editor

Marcio Rodrigues

Section Editor

Reviewer's Responses to Questions

**Key Review Criteria Required for Acceptance?**

**Methods**

-Are the objectives of the study clearly articulated with a clear testable hypothesis stated?

-Is the study design appropriate to address the stated objectives?

-Is the population clearly described and appropriate for the hypothesis being tested?

-Is the sample size sufficient to ensure adequate power to address the hypothesis being tested?

-Were correct statistical analysis used to support conclusions?

-Are there concerns about ethical or regulatory requirements being met?

Reviewer #1: Yes

Reviewer #2: The study aims evaluated the performance of molecular methods in the diagnosis of cryptococcal meningitis and was clearly stated. The design and population are adequate. This review is not sure the sample size is sufficient and recommend a statistician to review this issue. no concerns about ethical or regulatory requirements.

Reviewer #3: Methodologies listed were through and fulfilled necessary experimental control conditions. Comparison of the nested and multiplex PCR approaches to gold standard of CSF and Latex agglutination were key to interpretation whether these PCR from sera methodologies would be better or equal in detection of Cryptococcosis. Power analysis was not conducted, or at least not shown to determine if total of 49 patient samples were sufficient with individual groups ranging from n=18 to n=7 values. Groups being tests though are appropriate to address the hypothesis being tested at least with the group AIDS/HIV+ with cryptococcosis having the largest population size (n=18) for comparing across different methodologies. All ethical and regulatory requirements were conducted appropriately as the study was approved by the appropriate committees and patients signed appropriate consent forms.

**Results**

-Does the analysis presented match the analysis plan?

-Are the results clearly and completely presented?

-Are the figures (Tables, Images) of sufficient quality for clarity?

Reviewer #1: Yes

Reviewer #2: The analysis matched that planned. The results are clearly and completed presented, The figures are of quality.

Reviewer #3: The results section detailing the findings need to be supplemented with additional data showing replication of values projected as substantial in the accuracy and strength of agreement findings calculated by the inner rater agreement, Landis, and Koch-Kappa Benchmark Scale. Additionally, more clarification needs to be made on how the sensitivity and specificity differences were analyzed rather than a brief citation of previous methodologies.

**Conclusions**

-Are the conclusions supported by the data presented?

-Are the limitations of analysis clearly described?

-Do the authors discuss how these data can be helpful to advance our understanding of the topic under study?

-Is public health relevance addressed?

Reviewer #1: Yes

Reviewer #2: The conclusions are supported by the data presented, but the main concern is related to serum used to obtain colonies of Cryptococcus, which is a unreliable method for this aim. This review did not find the description os limitations of analysis, supposed to be written bu the authors. There are some discussion about the utility of the obtained data to addressed the cryptococcal meningitis diagnosis. The public health relevance was demonstred.

Reviewer #3: Public health relevance is addressed well here in this study but conclusions from the results again must be supported with additional supplemental data from the executed studies showing the values from the patients and their ranges deemed as substantial. Limitations of analysis are moderately clear.

**Editorial and Data Presentation Modifications?**

Reviewer #1: - There are no line numbers in the manuscript. This makes it difficult to track observations.

- In the context of identification/characterization, it is better to refer to Cryptococcus rather than cryptococcosis (throughout all the text). Although it is also usable.

Abstract

- In the introduction, it seems to me that a sentence is missing explaining the gap that the study intends to fill.

- Methodology >>> Methods

- The rapid India ink >>> The direct India ink staining (methods)

- Please provide comparative values for sensitivity and specificity of tests (results).

- The comparative aspect of the tests proposed as an objective of the manuscript is not clearly expressed in the conclusion.

Here a proposal: “Molecular techniques, in particular nested PCR specific to the 5.8Sr DNA sequence, show a better detection of Cryptococcus infection than conventional detection methods and 18S PCR, both in serum and CSF. They also allow a better early identification in case of negative serology and mycology, potentially allowing an early diagnosis”.

Introduction

- New data on the global burden of cryptococcosis in HIV patients were recently published in 2022 by Radha Rajasingham et al. https://doi.org/10.1016/ S1473-3099(22)00516-3. Please update

- Studies on systemic mycosis have shown that cryptococcosis is the second leading cause of AIDS-associated death from systemic mycosis in Brazil >>> after which opportunistic infection?

- The diagnosis of cryptococcosis is complex and is usually based on a combination of clinical and radiological suspicion followed by confirmation by laboratory diagnosis

- In patients with HIV-associated infections, culture of cerebrospinal fluid (CSF) is the gold standard for diagnosis of cryptococcal meningitis in viable sample…

Methods

- Although it is of little importance, I would not begin this section with ethical considerations.

- Serum and CSF from the same patient? Two types of samples per patient? Please specify –

- The study was approved by the institutional ethics committees of the Hospital das Clinicas da Faculdade de Medicina da Universidade de Sao Paulo - HCFMUSP (No. 0635/2007) and the Instituto de Infectologia Emílio Ribas - IIER (No. 63/2007).

- Patients >>> patients and samples

- After initial isolation in culture subsequent identification was by physiological and biochemical tests as described below: is it possible to quote at least the tests used?

- and AIDS/HIV-positive and positive for cryptococcosis (group IV; n=18) >>> and AIDS/HIV-positive and positive for cryptococcosis (group IV; n=18)

- (600-base pair [bp] product) and 18S 9429-bp product) >>> [600-base pair (bp) product] and 18S (9429-bp product).

- Multiplex PCR: Cryptococcus spp >>> Cryptococcus spp

- On the basis of the drawbacks of conventional routine techniques as notified in the introductory part, would you have the possibility to integrate the detection of inter-ribosomal DNA in the daily practice?

- The significance level was set at p ≤ 0.05: please make the font uniform.

Results

- How were non-cryptococcal agents identified in the samples? other fungi, bacteria and viruses?

- Table 2: Order of presentation of parameters = underlying diseases first, then HIV status.

- Prior to the Cryptococcus spp. assay, patient samples were subjected to nested PCR against the housekeeping gene GADPH, confirming in all samples the presence of human DNA and the absence of PCR inhibitors: This seems redundant to me because it has already been mentioned in the method section (page 10 - 11).

Also, the interpretive explanations of the Kappa test should be in the methods, they are not results. 

- All assays were performed in triplicate and showed 100% reproducibility. In addition, …

- It is essential to use the new nomenclature of Cryptococcus species: Cryptococcus var grubii >>> C. neoformans. This is for the whole manuscript

- The 18S PCR assay results in HIV-positive patients were similar for both sera and CSF, (Kappa index of 0.566 and 0552, respectively),

- However, when testing CSF, the 18S PCR exhibited a greater accuracy than did the latex agglutination test

- A comparative analysis of the 18SDNA primers with the latex agglutination assay for sera yielded a similar specificity and sensitivity, suggesting that the latex agglutination assay can be used as a complementary alternative test to this PCR test

- The comparative analysis between conventional (mycological India ink staining and serological)

- Overall, I note that the proportions compared by test are not presented. This could improve the understanding of some paragraphs.

Discussion

- The increased availability of anti-retroviral treatment (ART) plus aging of the HIV population may have contributed to changes in the spectrum of disease manifestation, including complications that may be attributable to the neurotoxic consequences of ART >>> I'm having trouble finding the connection of this sentence to the previous sentences.

- In addition, false-positive results in sera and CSF from HIV-positive and HIV- negative patients can occur under various conditions, such as in the presence of rheumatoid factors or infections with Trichosporon beigelii, Klebsiella pneumonia, Capnocytophaga canimorsus or Stomatococcus mucilaginosus. [20, 59].

- A comparative analysis of the 18SDNA primers with the latex agglutination assay for sera yielded a similar specificity and sensitivity, suggesting that the latex agglutination assay can be used as a complementary alternative test to this PCR test

- The lower sensitivity and specificity with the 18S primers in the CSF samples may be due to interference in gene amplification by components of the Cryptococcus spp. cell wall or the antiretroviral treatment >>> How do you explain the particularity of these two parameters at the level of 18S primers

- Multiplex PCR and sequencing data have confirmed that 95-100% of C. neoformans biological samples from AIDS patients worldwide, including in Brazil, are from the grubii strain (serotype A) >>> please consider the new nomenclature

- The present study has limitations that must be acknowledged. The patients’ medical records did not contain all relevant information, such as occupational risk factors. In addition, the analysis of patients’ follow-up was limited to their hospitalization and so the risk of recurrence or other complications after discharge could not be assessed and the small??? >>> truncated sentence?

- Although further studies are needed, our results indicate that the combination of nested 5.8S PCR testing in association with the latex agglutination test may increase the ability to diagnose and monitor cryptococcosis in patients.

Reviewer #2: none

Reviewer #3: Grammatical and sentence structured need to be fixed in the discussion and results sections. Some sentences are incomplete or run on to create confusing attempts at final conclusions. These are minor edits though that can be addressed quickly.

**Summary and General Comments**

Reviewer #1: The superiority of molecular techniques over conventional techniques is already established in the literature. However, the proposal of early diagnostic tests based on the analysis of the Cryptococcus genetic material seems to me more original and relevant. It is therefore preferable to put this aspect first. This will involve adaptations of some parts of the manuscript.

Reviewer #2: The objective of this study was to perform a comparative analysis of the sensitivity, specificity, and accuracy of conventional and molecular diagnostic methods for the detection of Cryptococcus neoformans in serum and cerebrospinal fluid (CSF) from individuals with meningitis. Although interesting, there are several concerns that need to be addressed with this manuscript.

The major concern is related to culture of serum to obtained Cryptococcus colonies (CSF and clotted blood samples were centrifuged….and the pellet fraction was cultured on SAB agar.) This is neither a conventional nor a recommended method. Blood cultures are preferred to serum for cultivation of fungi. 

 It is not clear the specimen used to obtain positive cultures is many sentences (e.g. Cryptococcus spp. were confirmed in 25 cases by isolation in culture or The 5.8S PCR exhibited the highest sensitivity and specificity with an accuracy of 100%; the Kappa index analysis showed almost perfect agreement with the gold standard (culture) in serum samples (0.918). This issue must be strictly defined because there are many possible interpretations of the results.

The authors should explain why they choose latex agglutination instead of lateral flow assay, the method used more extensively for immunological diagnosis of meningeal Cryptococcosis.

The English style deserves an extensive revision.

Additional comments:

Abstract:

The 5.8S DNA-ITS PCR yielded the highest levels of accuracy of any test? Or among the tested methods?

The correct sentence may be “negative serology and mycology results”.

Introduction section:

Please, revise the data because fungi are not a major contributor to opportunistic infections in those with advanced HIV infection.

The number of cases and death attributable to cryptococcal meningitis should be updated. A reference could be Rajasingham R, Govender NP, Jordan A, et al. The global burden of HIV-associated cryptococcal infection in adults in 2020: a modelling analysis [published online ahead of print, 2022 Aug 29] [published correction appears in Lancet Infect Dis. 2022 Oct 31;:]. Lancet Infect Dis. 2022;S1473-3099(22)00499-6. doi:10.1016/S1473-3099(22)00499-6 

The following information needs a reference “…indirect cases obtained from the STD/HIV-AIDS Department of the Brazilian Ministry of Health.”

Use initial capital letter for Cryptococcosis and other mycoses throughout the text.

The following reference on PCR-based diagnosis could be added: Martins Mdos A, Brighente KB, Matos TA, Vidal JE, Hipólito DD, Pereira-Chioccola VL. Molecular diagnosis of cryptococcal meningitis in cerebrospinal fluid: comparison of primer sets for Cryptococcus neoformans and Cryptococcus gattii species complex. Braz J Infect Dis. 2015;19(1):62-67. doi:10.1016/j.bjid.2014.09.004

Clarify the idea of “reducing patients’ risk of progression” for improve understanding.

Methods section:

Abbreviations and acronyms are often defined the first time they are used within the abstract and again in the main text and then used throughout the remainder of the manuscript. Please consider adhering to this convention (e.g. IIER and HCFMUSP).

There are also mistakes such as ml instead of mL and l instead 

of L (liter). Please, revise the entire manuscript.

It is not clear how many samples were collected 

from each patient.

Prefer Candida spp., instead of Candida sp.

Control strains section:

The entire paragraph should be reworded to improve understanding. 

Use square brackets for: H. capsulatum, ATCC 28308 (CDC: B973), ATCC 12700 (CDC: A811), and HC200 (GenBank: DQ239887) and aspergillosis (Aspergillus spp)

Mycology section:

Is it correct: clotted blood samples were centrifuged? What the purpose?

Nowadays, L-canavanine glycine bromothymol blue (CGB) medium is unsuitable for subtyped the species complexes of Cryptococcus. Molecular identification of the isolate is highly desirable.

Nested PCR section

Please, revise the marks: (600-base pair [bp] product) and 18S 9429-bp product). Also in Table 1 

Multiple PCR section

Use non-italics for spp. Revise the all text.

Sequencing Analysis section:

The sentence is repetitive: To avoid contamination of components, preparation of reaction mixtures and addition of template DNA were performed in separate rooms.

Results:

This entire paragraph can be eliminated as it was already presented in Methods section.

Regarding data presented in Table 2, the significance and impact on the Results have not been addressed. Otherwise, they are useless.

The sensitivity was 1 cell/ml for CSF, 10 cells/ml for serum and 1 cell/ml for culture isolates. These procedures for these points are not well explained in the Methods section.

The first letter of multiplex is low capter. 

This sentence does not make sense: Serum samples from two HIV-positive patients who were cryptococcal-negative yielded P. brasiliensis and H. capsulatum meningitis.

Would be good to clarify what “both groups” refers to in : Two samples from patients in both groups yielded false positive results for cryptococcosis, being instead positive for bacterial meningitis.

Table 3. Inform the number of samples used for each analysis

 The following methods were employed for Cryptococcosis diagnosis and not for identification of Cryptococcus spp. as informed: To further identify the preferred method for identification of Cryptococcus spp., we compared the results obtained by analysis of the 18S and 5.8S PCR-generated DNA sequences with results of latex agglutination and India ink staining. 

Discussion section:

The following paragraphs are out of scope: 

In our study of patients with meningitis, a diagnosis of cryptococcosis was confirmed in 51% of the cases, 72% of whom were HIV-positive and progressed to the disseminated form of cryptococcosis. This was consistent with prior findings [3, 11-12, 46-47] and agreed with the study of Yoon et al. [48]. The increased availability of anti-retroviral treatment (ART) plus aging of the HIV population may have contributed to changes in the spectrum of disease manifestation, including complications that may be attributable to the neurotoxic consequences of ART.

In both cryptococcosis positive and negative individuals, we observed a higher percentage of males in the cases with meningitis, according to Montoya et al., [49] one of the hypotheses may be hormonal, because testosterone increases the rate of melanin formation in C. neoformans cell and this may enhance pathogenesis [7, 36]

This review did not consider limitations of the study were absence of information on the occupational risk factors or the risk of recurrence or other complications after discharge.

Reviewer #3: Overall this paper attempts to present a novel utilization of using an established PCR methodology on a new biological sample as a means of non evasive and cost effective Cryptococcus infection diagnostic. While CSF testing and Latex agglutination are the gold standard methodologies at present, more non invasive and low cost methods are needed. The authors indeed at least from the table values presented, show support that 5.8SrDNA-ITS in combination with latex agglutination is a potential method to be utilized. The authors' notation of the limitations of this study highlight the awareness that more studies will need to be conducted in the future on a larger scale to confirm the use of nested PCR from serum as a viable diagnostic.

PLOS authors have the option to publish the peer review history of their article (what does this mean?). If published, this will include your full peer review and any attached files.

Reviewer #1: No

Reviewer #2: No

Reviewer #3: No
---

## [Decision Letter · Decision Letter 1]

5 Feb 2023

Dear Dr Dantas,

We are pleased to inform you that your manuscript 'Comparative analysis of diagnostic methods to detection Cryptococcus neoformans meningitis' has been provisionally accepted for publication in PLOS Neglected Tropical Diseases.

Best regards,

Chaoyang Xue, Ph.D.

Academic Editor

Marcio Rodrigues

Section Editor

Reviewer's Responses to Questions

**Key Review Criteria Required for Acceptance?**

**Methods**

-Are the objectives of the study clearly articulated with a clear testable hypothesis stated?

-Is the study design appropriate to address the stated objectives?

-Is the population clearly described and appropriate for the hypothesis being tested?

-Is the sample size sufficient to ensure adequate power to address the hypothesis being tested?

-Were correct statistical analysis used to support conclusions?

-Are there concerns about ethical or regulatory requirements being met?

Reviewer #1: - The study design is appropriate, with a correct study population.

- The objectives are well stated and structured. Just that the bridge between early detection of Cryptococcus in serum (and in CSF) and the cascade of complications that can occur outside of that detection is not explicitly mentioned in the rest the text.

- Small sample size but sufficient to test the hypothesis.

- Statistical analysis: sufficiently provided

- Regulatory requirements: met

Reviewer #3: yes

**Results**

-Does the analysis presented match the analysis plan?

-Are the results clearly and completely presented?

-Are the figures (Tables, Images) of sufficient quality for clarity?

Reviewer #1: Yes

Reviewer #3: yes

**Conclusions**

-Are the conclusions supported by the data presented?

-Are the limitations of analysis clearly described?

-Do the authors discuss how these data can be helpful to advance our understanding of the topic under study?

-Is public health relevance addressed?

Reviewer #1: - The conclusions are supported by data presented.

- Some limitations of the study are discussed.

- The contribution of the data presented to the advancement of our knowledge of cryptococcosis, as well as their impact on large-scale (policy) decision making, are little discussed, although all data are available.

Reviewer #3: yes

**Editorial and Data Presentation Modifications?**

Reviewer #1: Only minor modifications are necessary to improve the clarity of the data described in this manuscript.

Reviewer #3: yes

**Summary and General Comments**

Reviewer #1: - I still wonder about the use of serum as primary culture material, instead of preferring blood cultures.

- Overall, this is an interesting article that proposes the use of molecular techniques, which are very sensitive and specific, for the detection of Crypptococcus directly in a non-invasive sample, serum. This implies the legitimate link with the cost, time and unfortunate complications that can be due to the evolution of this disease.

Reviewer #3: This revised manuscript addressed reviewer comments effectively and included appropriate corrections.

PLOS authors have the option to publish the peer review history of their article (what does this mean?). If published, this will include your full peer review and any attached files.

Reviewer #1: No

Reviewer #3: No

---

## [Editor Report · Acceptance letter]

28 Feb 2023

Dear Dr Dantas,

We are delighted to inform you that your manuscript, "Comparative analysis of diagnostic methods to detection Cryptococcus neoformans meningitis," has been formally accepted for publication in PLOS Neglected Tropical Diseases.

Best regards,

Shaden Kamhawi

co-Editor-in-Chief

Paul Brindley

co-Editor-in-Chief
